

# Molecular sorting on a fluctuating membrane

Damiano Andreghetti[1,2], Luca Dall'Asta[1,2,3], Andrea Gamba[1,2,3]⋆,
Igor Kolokolov[4,5] and Vladimir Lebedev[4,5]

**1** Institute of Condensed Matter Physics and Complex Systems,
Department of Applied Science and Technology, Politecnico di Torino,
Corso Duca degli Abruzzi 24, 10129 Torino, Italy
**2** Istituto Nazionale di Fisica Nucleare (INFN), Via Pietro Giuria, 1, 10125 Torino, Italy
**3** Italian Institute for Genomic Medicine, Candiolo Cancer Institute,
Fondazione del Piemonte per l'Oncologia (FPO), Candiolo, 10060 Torino, Italy
**4** L.D. Landau Institute for Theoretical Physics, 142432, Moscow Region,
Chernogolovka, Ak. Semenova 1-A, Russia
**5** National Research University Higher School of Economics,
101000, Myasnitskaya 20, Moscow, Russia

⋆ andrea.gamba@polito.it

## Abstract

**Molecular sorting in biological membranes is essential for proper cellular function. It also plays a crucial role in the budding of enveloped viruses from host cells. We recently proposed that this process is driven by phase separation, where the formation and growth of sorting domains depend primarily on direct intermolecular interactions. In addition to these, Casimir-like forces—arising from entropic effects in fluctuating membranes —may also play a significant role in the molecular distillation process. Here, using a combination of theoretical analysis and numerical simulations, we explore how Casimir-like forces between rigid membrane inclusions contribute to sorting, particularly in the biologically relevant regime where direct intermolecular interactions are weak. Our results show that these forces enhance molecular distillation by reducing the critical radius for the formation of new sorting domains and facilitating the capture of molecules within these domains. We identify the relative rigidity of the membrane and supermolecular domains as a key parameter controlling molecular sorting efficiency, offering new insights into the physical principles underlying molecular sorting in biological systems.**

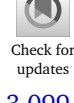

# 1  Introduction

Molecular sorting is a vital process in eukaryotic cells, where proteins and other biomolecules are sorted and encapsulated into lipid vesicles for targeted transport to specific subcellular locations. This distillation process occurs on lipid membranes, such as the plasma membrane [1], endosomes, the Golgi apparatus [2], and the endoplasmic reticulum [3], where biomolecules can bind and diffuse laterally. Due to a variety of direct and indirect interactions, these molecules aggregate into domains with distinct chemical compositions. These domains can induce membrane bending and fission [4–7], ultimately forming separated submicron lipid vesicles that are transported to their designated subcellular sites by molecular motors. In this way, lipid membranes act as natural molecular distillers, promoting intracellular order and compartmentalization and counteracting the homogenizing effects of diffusion. Disruption of molecular sorting in living cells is implicated in severe pathologies, including cancer [8, 9]. On the other end of the spectrum, analogous molecular sorting processes are exploited by enveloped viruses, such as HIV, SARS-CoV, and influenza, for their assembly and budding from host cells [10–13], further underscoring the practical relevance of understanding the physical mechanisms of molecular sorting.

We have recently proposed a simple model of molecular sorting as a phase-separation process. In this context, the efficiency of sorting is found to be optimal at intermediate values of intermolecular attraction forces [14–16]. This theoretical prediction is consistent with experiments on endocytic sorting in living cells under near-physiological conditions [14], and with measurements performed on photoactivated systems, where the strength of intermolecular attraction can be directly controlled [17]. The interpretation of molecular sorting as a phase-separation process is also coherent with the observation that sorting domains in living cells exhibit a critical size: only supercritical ("productive") domains evolve into lipid vesicles that are extracted from the membrane, while subcritical ("unproductive") domains are rapidly dissolved [15,18]. This perspective fits within the broader framework of the far-from equilibrium formation of biomolecular aggregates with specific functions, including lipid rafts [19,20] and specialized lipid-protein nanodomains [21–25], such as cadherin and integrin clusters [26,27].

Phase separation is emerging as one of the main ordering processes in living cells [28–30], and various mechanisms have been proposed as its drivers. Among them, weakly polar electrostatic interactions between disordered regions of proteins [31], active processes, as in diffusion-limited phase separation, mass-conserved reaction-diffusion systems and active emulsions [32–38], and segregating kinetic effects [39]. Moreover, it has long been established that protein inclusions in lipid membranes are subject to membrane-mediated interactions. These can originate either from ground state deformation of membrane shape, when protein inclusions are a source of intrinsic curvature, or from membrane fluctuations, as the presence

of embedded protein inclusions restricts membrane fluctuation modes, generating entropic interactions [40–42]. We focus here on the latter class of interactions, commonly known as Casimir-like forces. These are non-additive, weak forces that are mainly relevant at short separations [43–45]. At thermodynamic equilibrium, they are however sufficient to induce a demixing transition in heterogeneous membranes [46].

It is known that proteins and lipids involved in the formation of sorting domains increase local membrane rigidity by a factor of 10 to 30 compared to the surrounding membrane [47–49]. This suggests that entropic forces may play a relevant role in the molecular sorting process. Here we perform a thorough analysis of the problem and find that entropic forces significantly enhance molecular sorting efficiency, especially in the biologically relevant regime of weak direct interactions.

## 2 Phenomenological theory

Building on our previous work, we investigate the role of the lipid membrane as a distiller of molecular species [14–16]. In this scenario, molecules are randomly inserted into the membrane, diffuse laterally, and aggregate into sorting domains due to the action of attractive forces. The sorting domains grow by adsorbing molecules from the surrounding "gas" of freely diffusing molecules. Domains of size $R$ larger than a critical value $R_c$ grow irreversibly through the absorption of single molecules diffusing toward them [15, 50, 51]. The growth rate is determined by the net flux $\Phi$ of molecules toward a domain, which in turn is proportional to the molecular density difference $\Delta n = n_L - n_R$ between distant regions and regions adjacent to the domain boundaries [14]. Domains that reach a characteristic size $R_E$ are ultimately removed from the membrane through the formation of small, separate lipid vesicles [14]. It is worth observing here that vesicle formation is a complex process involving the concomitant action of a wide variety of genes, as reviewed, for instance, in Ref. [1]. In our approach, we abstract on molecular details and encode the mesoscopic effect of vesicle extraction in the single parameter $R_E$.

Of particular interest is the stationary out-of-equilibrium regime, where molecular insertion and extraction processes are balanced. This balance can be described by the equation

$$\phi = N_d \Phi, \tag{1}$$

where $\phi$ is the flux density of molecules being inserted into the membrane, $N_d$ is the density of supercritical domains, and $\Phi$ is the average flux of the molecules into a domain. In this regime, unlike in the classical Lifshitz-Slezov scenario [50, 51], the flux-driving jump $\Delta n$ in molecular density is kept finite by the continuous influx $\phi$ of molecules into the membrane.

We have shown in Ref. [14] that an optimal sorting regime is achieved for an intermediate strength of the attractive forces. When the tendency to aggregate is too strong, a proliferation of slowly growing sorting domains occurs, leading to molecular crowding and decreased sorting efficiency [14, 16]. In the optimal sorting regime, there exists a specific density $N_d$ of sorting domains, resulting in minimal average molecular density [14]. For absorbing domains, the average residence time $T$ of a molecule of linear size $a$ in the membrane system is the sum of the average time $T_f$ required for the molecule to reach a sorting domain by free diffusion and be absorbed, and the average time $T_d$ spent inside the domain until the extraction event. The two contributions can be estimated as [14]

$$T_f \sim \frac{1}{D N_d}, \qquad T_d \sim \frac{(R_E/a)^2}{\phi} N_d,$$

where $D$ is the molecular diffusion coefficient. The sum $T = T_f + T_d$ has a minimum for

$$N_{d,opt} \sim \frac{a}{R_E} \sqrt{\frac{\phi}{D}}. \tag{2}$$

The actual density $N_d$ is a function of the microscopic properties of the system that control the nucleation and growth of domains in the stationary state, but irrespective of the combination of these microscopic quantities, the optimal residence time of molecules on the membrane has the value determined by Eq. (2).

To account for the role of membrane fluctuations in the molecular sorting process described above, we recall that the equilibrium thermal fluctuations of an elastic membrane are described by the Helfrich Hamiltonian,

$$\mathcal{H} = \int dS \left[ \frac{\kappa}{2} \left( \frac{1}{R_1} + \frac{1}{R_2} \right)^2 + \frac{\bar{\kappa}}{R_1 R_2} \right], \tag{3}$$

where the integral runs over the membrane surface, $dS$ is the area element, $R_1, R_2$ are local principal curvature radii, and $\kappa, \bar{\kappa}$ are the bending rigidities associated with the mean and Gaussian curvatures, respectively [52–54]. As argued in Refs. [55–57], for biological membranes, $\bar{\kappa}$ is close to $-\kappa$. While our theory remains valid for any relation between $\kappa$ and $\bar{\kappa}$, for simplicity we will assume that $\bar{\kappa} = -\kappa$ in the numerical computations presented in the following section. In the presence of protein inclusions, the rigidity of the membrane becomes spatially non-uniform. Here, we assume that $\kappa(\mathbf{r}) = \kappa_0$ for the bulk membrane, and $\kappa(\mathbf{r}) = \kappa_1$ in the regions occupied by the molecules. A surface-tension contribution to the energy could also be included, but it is assumed to be negligible and will not be considered here.

We further assume that the diffusive dynamics of protein inclusions is slower than the fluctuational dynamics of the underlying membrane, i.e., $\tau_{diff} \gg \tau_{rel}$, with $\tau_{diff}$ the characteristic diffusion time and $\tau_{rel}$ the characteristic membrane relaxation time. This is motivated by the following estimates. The characteristic time for lateral diffusion can be estimated as $\tau_{diff} \sim \lambda^2/D$, where $\lambda$ is the characteristic scale of the problem. Assuming that the viscosity $\eta$ of the cytosol is the primary source of dissipation, the characteristic relaxation time of the membrane dynamics is $\tau_{rel} \sim \eta \lambda^3/\kappa$ [58]. Since the ratio $\tau_{rel}/\tau_{diff}$ increases as $\lambda$ grows, one should check whether the inequality $\tau_{diff} \gg \tau_{rel}$ holds for the largest characteristic scale, that is, for the size of the membrane. Considering membranes with sizes $\lambda = 100 - 500$ nm, taking the viscosity $\eta \sim 5 \cdot 10^{-3}$ Pa $\cdot$ s and the lateral diffusivity $D$ of proteins in the range $1 - 10 \ \mu m^2/s$ [59,60], one finds that the ratio $\tau_{diff}/\tau_{rel}$ spans the values $1 - 10^2$, suggesting that the dynamics of membrane fluctuations in living cells is faster than lateral particle diffusion [58,61,62].

## 2.1 Interaction with a sorting domain

Membrane fluctuations are known to induce effective interactions between inclusions within the membrane. These interactions can be conveniently studied in the weak fluctuation regime, where quantitative analyses can be performed [40,41,44,45,63,64]. It is of particular interest to investigate how these forces interplay with direct forces to facilitate the absorption of neighboring molecules by sorting domains. In the adiabatic approximation, justified by the timescale separation $\tau_{diff} \gg \tau_{rel}$, molecules included within the membrane experience effective forces that can be computed by averaging over membrane fluctuations sampled from the equilibrium distribution.

Analytic expressions for membrane-mediated forces can be derived in various limit cases. We are interested here in the interaction of a circular domain of size $R$ with a molecule of linear size $a$ situated at a distance $x$ from it. Approximating the domain boundary in zeroth

order as an infinite straight wall under the condition $R \gg x \gg a$, the effective potential energy of the membrane-mediated interactions is given by:

$$U(x) = -A\,k_{\mathrm{B}}T\frac{a^2}{x^2}\,, \tag{4}$$

where $A$ is a dimensionless, increasing function of the relative rigidity $\alpha = \kappa_1/\kappa_0$ (see Appendix A). Eq. (4) implies that $U \sim A\,k_{\mathrm{B}}T$ near the surface of a domain. On the other hand, the interaction potential between two inclusions mediated by the membrane fluctuations decays as $r^{-4}$ for distances $r$ much larger than their sizes [40]. Notice that when considering a membrane surface tension $\sigma$, a new relevant lengthscale, $\xi \sim \sqrt{\kappa/\sigma}$, emerges [65–67]. At scales below $\xi$, surface tension has a weak influence on membrane properties, whereas for scales above $\xi$, it significantly modifies the long-range part of the entropic interaction [44, 65–67]. As discussed in Refs. [44, 45] and in Appendix A, the entropic interaction is mainly appreciable at short separations. Therefore, we expect the effects of surface tension to be negligible in the present context.

## 2.2 Sorting process

The process of lateral diffusion of a molecule situated near a circular sorting domain can be described by the biased Brownian motion

$$\dot{\mathbf{r}} = -\beta D\nabla U(\mathbf{r}) + \xi\,,$$

where $\beta = (k_{\mathrm{B}}T)^{-1}$. According to the fluctuation-dissipation theorem, the noise term $\xi$ satisfies

$$\langle \xi_i(t) \rangle = 0\,,$$
$$\langle \xi_i(t)\xi_j(t) \rangle = 2D\delta_{ij}\delta(t-t')\,.$$

It is worth observing here that in the limit of weak fluctuations, geometric effects caused by the projection of the molecule's path can be neglected [68, 69]. Moreover, deviations of the domain shape from circularity produce rapidly decaying higher multipole contributions that may be neglected in the main approximation.

The time-dependent density profile of a population of such diffusing molecules around a domain obeys the following diffusion equation

$$\partial_t n(\mathbf{r},t) = \nabla \cdot [D(\nabla + \beta\nabla U)n(\mathbf{r},t)]\,, \tag{5}$$

where $n$ is the two-dimensional molecular density. To study the growth of the domain, one can consider an isotropic, time-independent solution to Eq. (5). The assumption of isotropy is justified by the circular shape of the domain, while the approximate time independence is supported by the slow nature of the diffusion process. Consequently, $n$ and $U$ depend only on the distance $r$ from the center of the domain. The explicit expression for $n(r)$ is given by:

$$n(r) = n(R)\exp[\beta U(R) - \beta U(r)] + \frac{\Phi}{2\pi D}\int_R^r \frac{\mathrm{d}\rho}{\rho}\exp[\beta U(\rho) - \beta U(r)]\,, \tag{6}$$

where $R$ is the radius of the domain and $n(R)$ is the molecular density near the domain boundary. For realistic values $\alpha \sim 10-30$ [47–49], the potential $U$, induced by membrane fluctuations, is at least of the order of $k_{\mathrm{B}}T$ when $r \sim R$ and tends to zero as $r$ grows (see Appendix A). The potential $U(r)$ rapidly approaches zero when $r$ becomes much larger than $R$ (as $\sim (r/R)^{-4}$,

see Eq. (A.14)). This allows us to neglect $U(r)$ in Eq. (6) when $r \gg R$. The leading contribution in $r/R$ can be extracted by integrating by parts in the integral in Eq. (6):

$$\mathcal{J} = \int_R^r \frac{\mathrm{d}\rho}{\rho} \mathrm{e}^{\beta U(\rho)} = \ln \frac{r}{R} + \delta \mathcal{J}, \tag{7}$$

where $\delta \mathcal{J}$ converges as $r \to \infty$:

$$\delta \mathcal{J} \approx -\beta \int_R^\infty \mathrm{d}\rho \frac{\mathrm{d}U(\rho)}{\mathrm{d}\rho} \ln\left(\frac{\rho}{R}\right) \mathrm{e}^{\beta U(\rho)}.$$

Since $|\delta \mathcal{J}| \leq (\pi A)^{1/2} a/R \ll 1$ for $a \ll R$, it can be neglected, leading to the relation

$$n(r) = n(R)\mathrm{e}^{\beta U(R)} + \frac{\Phi}{2\pi D} \ln \frac{r}{\tilde{R}}, \tag{8}$$

where $\tilde{R} \sim R$. For the attractive potential induced by membrane fluctuations, $U < 0$, $\tilde{R} > R$ and $\tilde{R} - R \sim R$. The factor $\mathrm{e}^{\beta U(R)}$ in Eq. (8) is of order unity.

The density of molecules near the domain boundary is determined by the dynamic equilibrium of association and dissociation processes and, using the Gibbs-Thomson relation [70,71], can be expressed as

$$n(R) = n_0(1 + R_\star/R), \tag{9}$$

where $n_0$ is the equilibrium density near a straight boundary, and the $R$-dependent correction accounts for the effect of linear tension. This correction is directly related to the curvature of the domain boundary. The length $R_\star$ in Eq. (9) can be estimated to be of the order of a few molecular radii. Expression (9) allows to determine the critical radius $R_c$: by definition, a domain with radius $R_c$ remains static, since the flux $\Phi$ for such a domain is zero. Substituting $\Phi = 0$ and $n(r) = n_L$ (where $n_L$ is the concentration of the molecules far from the domains) into Eq. (8) yields:

$$\frac{R_\star}{R_c} = \frac{n_L}{n_0} \mathrm{e}^{-\beta U} - 1. \tag{10}$$

Since $\exp(-\beta U) > 1$ for the attractive potential, we conclude from Eq. (10) that membrane-induced attraction reduces the critical radius. For domains larger than $R_c$, the correction related to the linear tension can be neglected, resulting in $n(R) \to n_0$. Consequently, we find from Eq. (8):

$$n_L - n_0 \mathrm{e}^{\beta U} = \frac{\Phi}{2\pi D} \ln \frac{L}{\tilde{R}}, \tag{11}$$

where $L$ is a distance of the order of the separation between the domains. Since $\exp(\beta U) < 1$ for the attractive potential, we conclude from Eq. (11) that membrane-mediated attraction enhances the effectiveness of the clustering process, resulting in an increased flux $\Phi$.

The above relations show how forces mediated by membrane fluctuations affect the sorting process. Let us examine the effect of increasing membrane-mediated attraction (which can be directly adjusted in numerical simulations by varying the relative rigidity $\alpha = \kappa_1/\kappa_0$). As membrane-mediated attraction increases, the critical radius $R_c$ of the domains decreases, leading to a higher rate of production of germs of new sorting domains and, consequently, an increased overall density $N_d$ of sorting domains [51]. However, according to the balance relation (1), this should concomitantly result in a lower $\Phi$ and, in accordance with Eq. (11), a lower $n_L$, which in turn reduces the rate of new domain generation. Between these two opposing effects, the first is expected to dominate due to the high sensitivity of the process of germ generation to the critical radius $R_c$ [51]. The following physical picture thus emerges. The efficiency of the sorting process is controlled by the rate of nucleation of new sorting domains. According to classical nucleation theory the rate of generation of new sorting domains

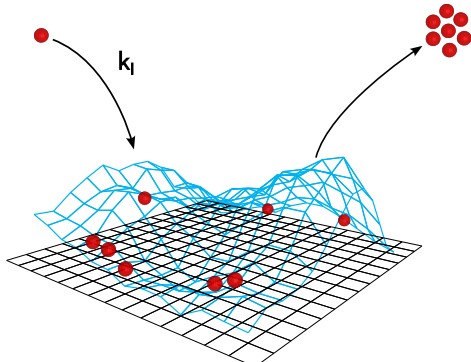

Figure 1: Schematic representation of the discrete model of molecular sorting on a fluctuating membrane. The membrane (in blue) is described by its height relative to a reference plane (in black). Rigid molecules are inserted into vacant sites at a rate $k_I$, and connected domains containing more molecules than the threshold size $N_E$ are extracted. The amplitude of membrane fluctuations is here amplified for clarity.

depends exponentially on $R_c$ [51]. Eq. (10) implies that even entropic interactions $\beta U \sim 1$ significantly affect $R_c$. At short distances, the entropic force acts as a facilitator of nucleation by biasing molecular diffusion toward sorting domains and stabilizing them. While at equilibrium a sharp demixing transition is observed above a critical value of rigidity [46], in the statistical steady state of interest here we expect to observe a smooth increase of the rate of nucleation of sorting domains with increasing rigidity of the inclusions. A significant effect is expected in particular in the realistic range $\alpha \sim 10 - 30$ [47–49], where $\beta U \sim 1$ in the proximity of the domains.

It is worth observing here that in the statistical steady state, the density $N_d$ of sorting domains is self-consistently determined through the stationarity condition $\mathrm{d}N_d/\mathrm{d}t = \phi/N_E$, where $N_E$ is the average number of molecules removed during an extraction event, since the rate of formation of new domains $\mathrm{d}N_d/\mathrm{d}t$ is in average equal to the rate of extraction events. [14, 15]. Starting from the regime of weak direct interactions, the optimal density of sorting domains $N_{d,\mathrm{opt}}$ determined by Eq. (2) can be reached either by increasing the direct interaction strength, or by reducing the critical radius by means of increased molecular rigidities $\kappa_1$. Conversely, to increased molecular rigidities should correspond lower values of the optimal direct interaction strength.

## 3 Numerical results

To validate our theoretical predictions, we implemented a numerical scheme that generalizes the lattice-gas model of molecular sorting introduced in Ref. [14]. This scheme shares several features with the approach used in Ref. [46] to investigate the phase separation of rigid inclusions in fluid membranes close to thermodynamic equilibrium, although we are studying here an out-of-equilibrium state. We consider a fluctuating membrane described by a discretized version of Helfrich Hamiltonian, on which inserted molecules laterally diffuse and aggregate. The system is driven out of equilibrium by an incoming flux of molecules, which are randomly attached at empty membrane sites with a rate $\phi$ per unit area, and is maintained in a statistical stationary state by the instantaneous removal of connected molecular domains that reach the threshold number of molecules $N_E$. Consistently with our theoretical approach, simulations are performed in the adiabatic regime.

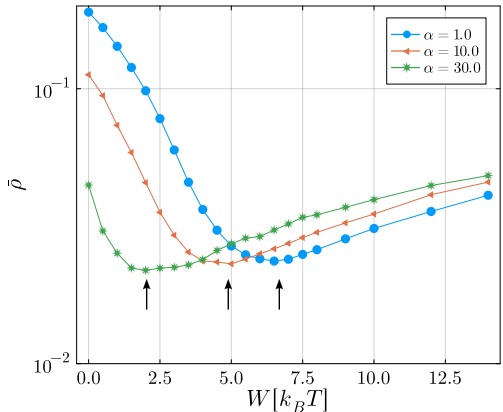

Figure 2: Average density $\bar{\rho}$ in the stationary state as a function of the direct inter-action strength $W$. The different curves correspond to different values of $\alpha = \kappa_1/\kappa_0$. The optimal sorting region depends on both the direct interaction and the rigidity of the biomolecules involved. For larger values of the relative rigidity $\alpha$, the density curve and the optimal interaction strength $W_{\text{opt}}$ shift toward lower values of $W$. Simulations were performed with $\phi/k_D = 10^{-5}$, $\kappa_0 = -\bar{\kappa}_0 = 10\,k_B T$, $\kappa_1 = -\bar{\kappa}_1$, $L = 100$, $N_E = 25$. For $h = 10\,\text{nm}$ and $D = 1\,\mu\text{m}^2/\text{s}$, one has $k_D^{-1} = 10^{-4}\,\text{s}$.

In our numerical scheme, the membrane configuration is described by the height $u_i$ of its points relative to a reference plane, which is discretized into a square lattice of $L \times L$ sites, see Fig. 1. To avoid boundary effects, periodic boundaries conditions are applied. Each site of the lattice can be occupied by at most one molecule. An occupation number $n_i \in \{0, 1\}$ is associated to each site $i$. Sites with $n_i = 0$ have the bending rigidity $\kappa_0$, while sites with $n_i = 1$ have the rigidity $\kappa_1$. The corresponding Gaussian rigidities are assumed to be equal to $-\kappa_0$ and $-\kappa_1$, respectively. To account for the direct attractive force between membrane inclusions we add to the discretized Helfrich energy of the membrane the nearest-neighbor interaction energy

$$H_{\text{incl}} = -\frac{W}{2} \sum_{\langle i,j \rangle} n_i n_j \,. \tag{12}$$

Membrane configurations are sampled using a Monte Carlo algorithm. After each Monte Carlo sweep (MCS), steps involving molecule insertion, diffusion, and the extraction of domains of size $\geq N_E$ are performed. One MCS is taken as the time unit. The rate of molecule insertion per empty site is denoted by $k_I$. The diffusion rate $k_D$ of free molecules is measured as the ratio of accepted diffusive jumps during one MCS (see Appendix B for additional details). Simulations are performed with the realistic parameter values $\kappa_0 = 10\,k_B T$, $N_E = 25$ [14–16,72,73], while $k_I$ and $k_D$ are kept much smaller than 1 in inverse MCS units, to ensure proper sampling of membrane configurations within the adiabatic regime. To match simulation parameters with real-world units, we consider that each square plaquette in the lattice corresponds to a patch of lipids of area $h^2$, with $h \approx 10\,\text{nm}$, the order of magnitude of the lateral size of typical protein inclusions, and also of the shortest fluctuational wavelengths for membrane bending deformations [46,74]. For molecular diffusivities $D \approx 1\,\mu\text{m}^2/\text{s}$, the typical time between consecutive diffusive jumps of a free inclusion on the lattice is $k_D^{-1} = h^2/D \approx 10^{-4}\,\text{s}$.

The average density $\bar{\rho}$ of molecules in the stationary state satisfies the relation $\bar{\rho} = \phi\,T$, where $T$ is the average time a particle spends on the membrane before being extracted, and $\phi = k_I(1 - \rho)$ is the flux of incoming particles per site, if lengths are measured in units of the lattice spacing [14,75]. Therefore, in the statistically stationary state established at fixed $\phi$, the average density $\bar{\rho}$ is a measure of the efficiency of the sorting process [14].

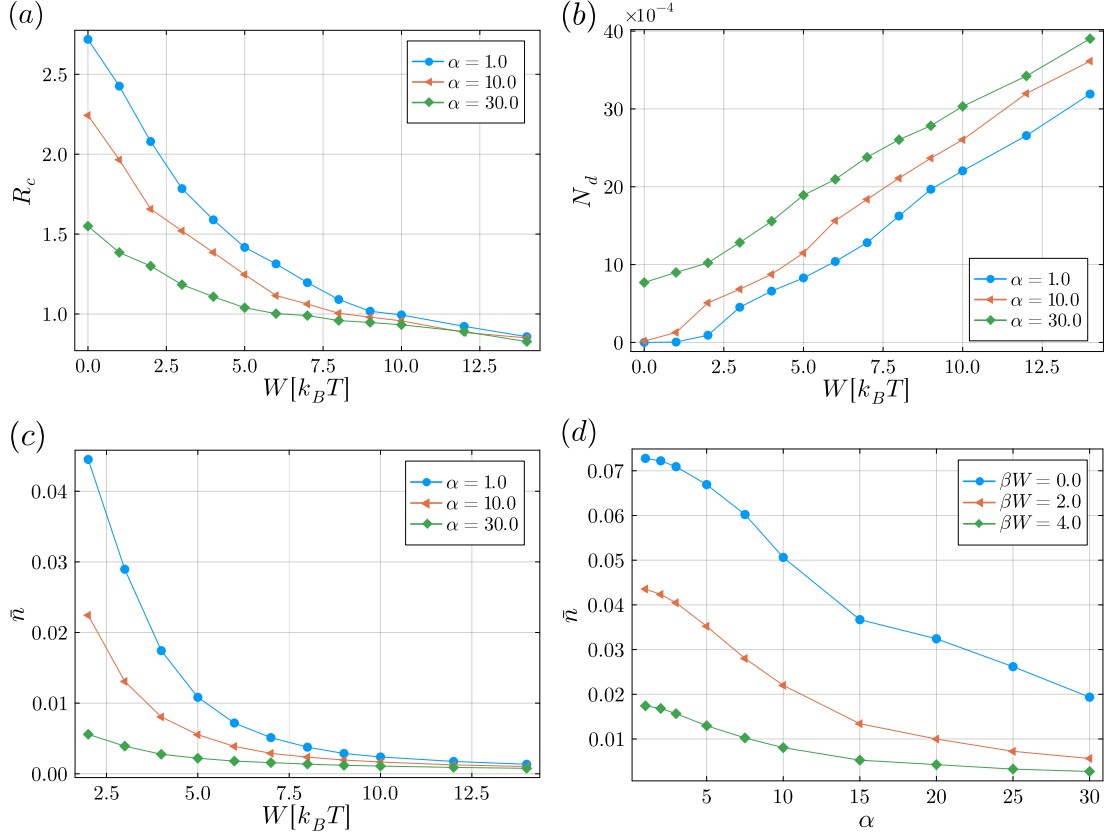

Figure 3: Characterization of the sorting process in the statistically steady state in terms of three key observables, measured from numerical simulations as functions of the direct interaction strength $W$, for varying relative rigidities $\alpha = \kappa_1/\kappa_0$: (a) the critical radius $R_c$ (estimated using the method described in Ref. [15]); (b) the number density $N_d$ of supercritical domains; (c) the average density of isolated molecules $\bar{n}$; and (d) the average density of isolated molecules $\bar{n}$ as a function of $\alpha$ for three different values of $W$. Due to the logarithmic profile of the molecular density around sorting domains, the average density $\bar{n}$ is close to $n_L$. Same parameters as in Fig. 2.

We investigated the behavior of the density $\bar{\rho}$ as a function of the direct interaction $W$ and molecular rigidity $\kappa_1$. In Fig. 2, the resulting stationary densities are plotted as functions of the direct interaction strength $W$ for the fixed dimensionless flux $\phi/k_D = 10^{-5}$ (see Appendix B), with varying $\alpha = \kappa_1/\kappa_0$.

These numerical results confirm the theoretical prediction that membrane-mediated interactions strongly influence the molecular sorting process, and that the optimal direct interaction strength $W_{\text{opt}}$ decreases as the intensity of membrane-mediated interactions increases (Fig. 2), thus enhancing sorting efficiency in the biologically relevant regime of weak direct interactions. Since the entropic force mainly acts at short separations, it renormalizes the value of the direct interaction, resulting in an effective short-range interaction strength $W_{\text{eff}}$, as evidenced by the consistent shift of the density curves toward lower values of $W$ in Fig. 2. However, it is important to note that the entropic component of $W_{\text{eff}}$ has a distinct origin and parametric dependence compared to the direct interaction part, as it is governed by the relative rigidity $\alpha$.

To further validate the present theoretical scenario, we measured the critical size $R_c$, the number density of sorting domains $N_d$, and the average density of isolated molecules $\bar{n}$ (which

is approximately the same as $n_L$) for varying values of $W$ and $\alpha$ (Fig. 3). Consistent with the theoretical predictions, the critical size $R_c$ decreases monotonically with both increasing $W$ and $\alpha$ (Fig. 3a), resulting in a higher sorting domain density $N_d$ (Fig. 3b). This confirms that, in the presence of membrane-mediated interactions, the optimal sorting-domain density $N_{d,opt}$ is achieved at lower direct interaction strengths $W$. As predicted, the increase in sorting-domain density is reflected in a corresponding decrease in the average density of isolated molecules $\bar{n}$ (Fig. 3c). As expected, the effect of entropic forces on the sorting process exhibits a smooth dependence on the relative rigidity $\alpha$, and it becomes particularly significant for $\alpha \gtrsim 10$, where $\beta U \sim 1$ (Fig. 3d). Furthermore, we tested the effect of including a surface term in our simulations and observed no significant change up to the realistic value $\sigma = 10^{-5}\,\mathrm{J\,m^{-2}}$ [74], in line with theoretical arguments.

## 4  Conclusions

The lipid membranes of endosomes, the Golgi apparatus, the endoplasmic reticulum, and the plasma membrane play a fundamental role in sorting and distilling vital molecular factors, acting as a natural realization of Szilard's model of classical nucleation theory [51]. These delicate structures are inherently subject to thermally induced fluctuations. Previous studies have shown that such fluctuations significantly contribute to the phase separation of rigid membrane inclusions close to thermodynamic equilibrium [46]. Our analysis extends these findings to the out-of-equilibrium scenario of molecular sorting, demonstrating that membrane-mediated interactions can strongly enhance the molecular distillation of rigid inclusions, particularly, in the biologically relevant regimes where direct intermolecular attractive forces are relatively weak. Our analysis suggests that thanks to membrane-mediated interactions, rigid biomolecules can be sorted with high efficiency, despite their low-affinity interactions. Notably, this effect, potentially crucial for biological systems, is observed in our numerical simulations well below the threshold where phase separation occurs close to equilibrium [46]. This suggests an important distinction between classical quasi-equilibrium phase separation processes and the role phase separation plays in out-of-equilibrium biological systems. Note that from the point of view of macroscopic kinetics, the entropic forces can be described by a short-range interaction of the same type as in Eq. (12), with an effective parameter $W_{eff}$, corresponding to attraction of particles toward the domains. The value of $W_{eff}$ is determined by the relative rigidity of the domain and has order $k_B T$. As a result, at low and moderate values of $W$, entropic forces provide a universal physical mechanism for the aggregation of molecules within the membrane, independent of microscopic interaction details. We believe that this should be taken into account in the design and interpretation of biological experiments.

Molecular inclusions interact with the surrounding membrane due to both their rigidity and, possibly, non-zero intrinsic curvature [40, 76]. In this study, we have focused on the impact of rigidity on the molecular sorting process. In future work, we plan to investigate the complex interplay between rigidity and intrinsic curvature.

Here, we did not account for the role of the cytoskeleton, which could influence the membrane fluctuation spectrum. At the mesoscopic scale, a more accurate description can be achieved by incorporating additional terms into the membrane Hamiltonian (Eq. 3) to account for interactions with the cytoskeletal network. Previous research has explored this aspect by introducing local membrane pinning [77, 78] or by considering membrane confinement [79]. These studies found that at short wavelengths, the fluctuation spectrum of a free membrane is retrieved. This suggests that the effect of the cytoskeleton on entropic forces, which in the present context mainly act at short separations, may be weak. This point will be the subject of future investigations.

Our findings suggest that a key parameter governing molecular sorting efficiency is the relative rigidity of the membrane and supermolecular domains, which affects the critical radius for the nucleation of nascent sorting domains. The statistical distribution of domain sizes provides an accessible signature of many self-organized aggregation processes [24, 26]. The domain size distribution predicted by our model [14] aligns well with experimental data for endocytic sorting and can be used to infer the critical size [15]. This suggests a practical way to experimentally investigate the physical picture of molecular sorting proposed in this work.

## Acknowledgments

AG would like to thank Guido Serini for many fruitful discussions.

**Funding information**  Numerical calculations have been made possible through a CINECA-INFN agreement providing access to computational resources at CINECA.

## A  Interaction of a molecule with a domain

In this section, we analyze the Casimir interaction between a circular domain of radius $R$ and a single molecule of radius $a \ll R$, positioned at a distance $x \gg a$ from it. We will calculate the interaction potential between the molecule and the domain.

In the absence of overhangs, the membrane can be parameterized in the Monge gauge [80], where each point on the membrane is defined by its displacement $u(\mathbf{r}) = u(x, y)$ in the direction perpendicular to a reference plane $\mathcal{S}$. To second order in $u$, the Helfrich Hamiltonian, which provides the elastic energy of the deformed membrane, reads

$$\mathcal{H} = \int_{\mathcal{S}} \mathrm{d}x\, \mathrm{d}y \left\{ \frac{\kappa}{2} (\nabla^2 u)^2 + \bar{\kappa}[\partial_x^2 u\, \partial_y^2 u - (\partial_x \partial_y u)^2] \right\}, \tag{A.1}$$

Here $\kappa$ and $\bar{\kappa}$ are bending and Gaussian rigidities, determined by an internal structure of the membrane. A surface-tension contribution to the energy could also be included, but it is assumed to be negligible and will not be taken into account.

Here we consider the interaction of a single molecule with a circular domain of molecules inserted into the membrane. When the molecule is positioned at the point $\mathbf{r} = (x, y)$, the interaction potential of the molecule with the domain is

$$\begin{aligned} U = \; & B(\partial_x^2 \partial_{x'}^2 \mathcal{G}|_{x=x', y=y'} + 2\partial_x^2 \partial_{y'}^2 \mathcal{G}|_{x=x', y=y'} + \partial_y^2 \partial_{y'}^2 \mathcal{G}|_{x=x', y=y'}) \\ & + D(\partial_x^2 \partial_{y'}^2 \mathcal{G}|_{x=x', y=y'} - \partial_x \partial_y \partial_{x'} \partial_{y'} \mathcal{G}|_{x=x', y=y'}), \end{aligned} \tag{A.2}$$

where $\mathcal{G}(\mathbf{r}, \mathbf{r}')$ is the contribution to the pair correlation function $\langle u(\mathbf{r}) u(\mathbf{r}') \rangle$ from the membrane displacement induced by the domain. The factors $B, D$ in Eq. (A.2) are introduced via the phenomenological coupling energy of the molecule with the membrane, when the former is treated as a point-like object:

$$\delta\mathcal{H} = B(\nabla^2 u)^2 + D[\partial_x^2 u \partial_y^2 u - (\partial_x \partial_y u)^2], \tag{A.3}$$

where the derivatives are evaluated at the position of the molecule. This expression is valid for fluctuations of $u$ on scales much larger than $a$. The factors $B$ and $D$ are functions of the rigidity and size of the molecule. We will make use of the fact that their expression for a disc

of radius $a$ and rigidity $\kappa = \kappa_2$, $\bar\kappa = -\kappa_2$, inserted in a membrane of rigidity $\kappa = \kappa_0$, $\bar\kappa = -\kappa_0$ is [44, 66]:

$$
\begin{aligned}
B &= \pi a^2 \kappa_0 (\kappa_2 - \kappa_0) \left( \frac{1}{(\kappa_2 + \kappa_0)} + \frac{1}{\kappa_2 + 3\kappa_0} \right), \\
D &= -\pi a^2 \frac{4(\kappa_2 - \kappa_0)\kappa_0}{\kappa_2 + 3\kappa_0}.
\end{aligned}
\tag{A.4}
$$

If the separation between the molecule and the domain boundary is much smaller than the domain size $R$, the boundary can be approximated as a straight line. Therefore, we assume that the domain occupies the half-plane $x < 0$. We also consider that the domain and the bulk membrane have different bending and Gaussian rigidities, $\kappa_1, \bar\kappa_1$ and $\kappa_0, \bar\kappa_0$. respectively. The Hamiltonian of the system is then given by

$$
\begin{aligned}
\mathcal{H} = &\int_{\mathcal{D}_1} dx\, dy\, \left\{ \frac{\kappa_1}{2}(\nabla^2 u)^2 + \bar\kappa_1 [\partial_x^2 u \partial_y^2 u - (\partial_x \partial_y u)^2] \right\} \\
&+ \int_{\mathcal{D}_2} dx\, dy\, \left\{ \frac{\kappa_0}{2}(\nabla^2 u)^2 + \bar\kappa_0 [\partial_x^2 u \partial_y^2 u - (\partial_x \partial_y u)^2] \right\},
\end{aligned}
\tag{A.5}
$$

where $\mathcal{D}_1$ is the left half-plane ($x < 0$) and $\mathcal{D}_2$ is the right half-plane ($x > 0$).

Using linear response theory, we can derive an equation for the pair correlation function $G = \langle u(\mathbf{r})u(\mathbf{r}') \rangle$, entering Eq. (A.2). It is important to note here that, due to the system's homogeneity in the $y$ direction and its invariance under reflection $y \to -y$, $G$ is a function of $|y - y'|$. The resulting equations read

$$
\begin{aligned}
\nabla^4 G &= \frac{k_\mathrm{B}T}{\kappa_1}\delta(x - x')\delta(y - y'), \qquad x < 0, \\
\nabla^4 G &= \frac{k_\mathrm{B}T}{\kappa_0}\delta(x - x')\delta(y - y'), \qquad x > 0,
\end{aligned}
\tag{A.6}
$$

with boundary conditions

$$
\begin{aligned}
\partial_x(\kappa_1 \nabla^2 - \bar\kappa_1 \partial_y^2)G|_{x=0^-} &= \partial_x(\kappa_0 \nabla^2 - \bar\kappa_0 \partial_y^2)G|_{x=0^+}, \\
(\kappa_1 \nabla^2 + \bar\kappa_1 \partial_y^2)G|_{x=0^-} &= (\kappa_0 \nabla^2 + \bar\kappa_0 \partial_y^2)G|_{x=0^+}.
\end{aligned}
\tag{A.7}
$$

Observe that, due to the inhomogeneity of the Gaussian rigidity, the topological term involving Gaussian curvature in the Hamiltonian cannot be neglected. This term contributes to the boundary conditions (A.7) for the correlation function.

Due to translation invariance along the $y$ direction, it is convenient to make use of the Fourier transform

$$
\hat{G}(x, x', q) = \int_{-\infty}^{+\infty} dy\, \exp[iq(y - y')]G(x, x', y - y'),
$$

which is an even function of $q$. The solutions to Eqs. (A.6) and (A.7) for $q > 0$ are

$$
\hat{G}(x, x', q) = (A_0 + A_1 x)\,e^{qx} + \frac{k_\mathrm{B}T}{4q^3 \kappa_1}(1 + q|x - x'|)\,e^{-q|x-x'|},
$$

for $x < 0$, and

$$
\hat{G}(x, x', q) = (B_0 + B_1 x)\,e^{-qx} + \frac{k_\mathrm{B}T}{4q^3 \kappa_0}(1 + q|x - x'|)\,e^{-q|x-x'|},
$$

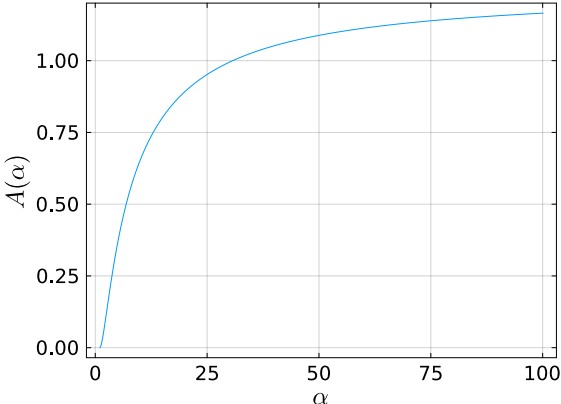

Figure 4: Dependence of the prefactor $A$ from Eq. A.11 on the relative rigidity $\alpha$.

for $x > 0$. The factors $A_0, A_1, B_0, B_1$ must be determined from the continuity of $\hat{G}$ and its derivative $\partial_x \hat{G}$ at $x = 0$ and from the boundary conditions (A.7), where $\partial_r^2 \to -q^2$, $\nabla^2 \to \partial_x^2 - q^2$. Assuming $\bar{\kappa}_0 = -\kappa_0$ and $\bar{\kappa}_1 = -\kappa_1$, the correlation function for $x, x' > 0$ is

$$\hat{G}(x, x', q) = \frac{k_B T}{4q^3 \kappa_0} \Bigg[ (1 + q|x - x'|) e^{-q|x-x'|}$$
$$- \frac{e^{-q(x+x')}(\kappa_1 - \kappa_0)((3\kappa_1 + \kappa_0)(x + x' + 2qxx')q + 3\kappa_1 + 5\kappa_0)}{(3\kappa_1 + \kappa_0)(\kappa_1 + 3\kappa_0)} \Bigg]. \quad \text{(A.8)}$$

The second term in the square brackets determines the contribution $\mathcal{G}$ to the correlation function induced by the domain.

In accordance with Eqs. (A.2, A.4, A.8) the interaction energy of the molecule with the domain is

$$U(x) = -k_B T \frac{(\kappa_1 - \kappa_0)}{4(\kappa_1 + 3\kappa_0)} \left( \frac{\kappa_2 - \kappa_0}{\kappa_2 + 3\kappa_0} \right) \left[ \frac{15\kappa_2 \kappa_1 + 13\kappa_2 \kappa_0 + 21\kappa_0 \kappa_1 + 15\kappa_0^2}{(\kappa_2 + \kappa_0)(\kappa_0 + 3\kappa_1)} \right] \frac{a^2}{x^2}. \quad \text{(A.9)}$$

When the molecule and the domain have the same rigidity ($\kappa_2 = \kappa_1$), we obtain

$$U(x) = -A k_B T \frac{a^2}{x^2}, \quad \text{(A.10)}$$

where, letting $\alpha = \kappa_1 / \kappa_0$,

$$A = \frac{(\alpha - 1)^2 (3\alpha + 5)(5\alpha + 3)}{4(\alpha + 1)(\alpha + 3)^2 (3\alpha + 1)}, \quad \text{(A.11)}$$

which is a monotonically increasing function for $\alpha > 1$, taking on values of order 1 for $\alpha \gtrsim 10$ (Fig. 4).

**Interaction at large distances** At large separations between the molecule and the domain, the size $R$ of the domain becomes a relevant scale, and its boundary can no longer be treated as an infinite wall. In this case, the interaction can be evaluated as [66]

$$U(x) = \frac{BD_R + B_R D}{2\pi^2 \kappa_0^2 x^4} k_B T = -\tilde{A} k_B T \frac{a^2 R^2}{x^4}, \quad \text{(A.12)}$$

where

$$\tilde{A} = \frac{2(\alpha - 1)^2 (3\alpha + 5)}{(\alpha + 1)(\alpha + 3)^2}. \quad \text{(A.13)}$$

Note that, by taking the appropriate limits, this expression reproduces previous analytical results found in the literature [40, 44].

When considering a single molecule diffusing in the vicinity of a sorting domain, one of the two regimes in Eq. (A.10) and Eq. (A.12) should be considered depending on the distance. A convenient interpolation formula for the membrane-mediated interaction energy between a molecule and a sorting domain of radius $R$, valid across different asymptotic regimes, is given by the simplest two-point Padé approximant [81]

$$U(r) = -k_{\mathrm{B}}T \frac{R^2}{r^2} \left[ \frac{A a^2}{(r-R)^2 + a^2} + (\tilde{A}-A)\frac{a^2}{r^2} \right], \qquad (A.14)$$

where $r = x + R$ is the distance from the molecule to the center of the domain. This reduces to Eq. A.12 when $r \gg R$, $r \gg a$, and to Eq. A.10 in the limit $r \sim R$ and $r - R \gg a$, while also avoiding the unphysical singularity at $x = 0$.

**Interaction in the proximity of the domain**    In the previous paragraphs, we assumed that the distance $x$ between molecule and domain was much larger than the size $a$ of the molecule. Here, we take into account the finite size of the inclusion by modeling it as a disk of rigidity $\kappa_2$ and radius $a$ centered in the point of coordinates $(x, 0)$:

$$\delta\mathcal{H}_{\mathrm{Disk}} = \frac{(\kappa_2 - \kappa_0)}{2} \int_{\mathrm{Disk}} d^2\mathbf{r} \left\{ (\nabla^2 u(\mathbf{r}))^2 - 2[\partial_x^2 u(\mathbf{r})\partial_y^2 u(\mathbf{r}) - \partial_x\partial_y u(\mathbf{r})\partial_x\partial_y u(\mathbf{r})] \right\} . \quad (A.15)$$

The resulting interaction energy can be computed as

$$U_{\mathrm{Disk}} = -k_{\mathrm{B}}T \log \frac{Z}{Z_0} = -k_{\mathrm{B}}T \sum_{n=1}^{\infty} \frac{(-\beta)^n}{n!} \langle \{\delta\mathcal{H}_{\mathrm{Disk}}\}^n \rangle_{0,\mathrm{c}} , \qquad (A.16)$$

where $\langle \cdots \rangle_{\mathrm{c}}$ denotes connected averages. Resummation of similar perturbation series has been performed up to finite orders in analogous geometries, either through direct computation of Feynman diagrams [45] or via numerical methods [44], demonstrating that the interaction energy increases sharply at short separations. We have evaluated both $U$ from Eq. A.9 and $U_{\mathrm{Disk}}$ from Eq. A.16 at first order in $(\kappa_2 - \kappa_0)/\kappa_0$, getting

$$\frac{U_{\mathrm{Disk}}^{(1)}}{U^{(1)}} = 2\left[ \frac{1}{\sqrt{1-(a/x)^2}} - 1 \right] \left(\frac{x}{a}\right)^2 .$$

This relative finite-size correction is plotted in Fig. 5. It shows that Eq. A.9, which neglects finite-size effects, significantly underestimates the interaction energy at short distances while accurately capturing its behavior at separations larger than the inclusion size.

## B    Simulation protocol

Simulations are performed according to a protocol that employs a Monte Carlo technique to sample Gibbs distributed configurations of the membrane, and a sub-lattice continuum Langevin equation for particle dynamics within lattice cells. Each Monte Carlo sweep (MCS) is executed as follows:

**Membrane:**    Each site of the lattice is visited in random order, and a random displacement of the height of the surface at that site is proposed, with uniform probability within an interval of amplitude $2l_0$ centered around the previous position. The move is accepted or rejected according to the Metropolis criterion. The value of $l_0$ is chosen to achieve an acceptance rate of approximately 50% for the proposed moves.

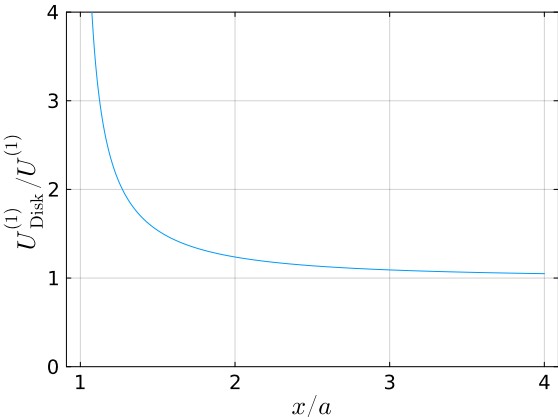

Figure 5: Finite-size correction to the interaction energy. The curve shows the ratio between the interaction energy calculated using Eq. A.16 to that from Eq. A.9, with both expressions expanded to first order in $(\kappa_2 - \kappa_0)/\kappa_0$. This ratio is plotted as a function of the ratio of the distance from the wall, $x$, normalized by the lateral inclusion size, $a$. The correction is especially significant at short distances.

**Diffusion:** After each membrane MCS, each lattice site $i$ is visited in random order. If a particle is present, the auxiliary variables $x_i^{(t)}$ and $y_i^{(t)}$ are updated according to the following rule:

$$
\begin{aligned}
x_i^{t+1} &= x_i^t + \frac{F_x^t(x_i^t) + \sqrt{2\gamma k_\mathrm{B} T}\,\eta^t}{\gamma}\,, \\
y_i^{t+1} &= y_i^t + \frac{F_y^t(y_i^t) + \sqrt{2\gamma k_\mathrm{B} T}\,\eta^t}{\gamma}\,,
\end{aligned}
\tag{B.1}
$$

where $\eta^t$ is a Gaussian noise with zero mean and variance 1, and $F_x^t(x), F_y^t(y)$ are forces acting on the molecule along the $x$ and $y$ directions at time t and position $(x, y)$. The constant $\gamma$ plays the role of the friction coefficient in the Langevin equation and sets the average length of the discrete steps of the auxiliary random walk. To ensure effective sampling, it is required that $\gamma \gg |F|$. The coordinates $(x_i^{(t)}, y_i^{(t)})$ can be interpreted as the sublattice position of the molecule at site $i$ at time $t$. The forces acting on the particle are evaluated as $-\nabla U$, where $U$ is the discretized membrane energy, smoothed through a quadratic interpolation, in order to achieve sub-lattice resolution. When reaching the jump condition $x_i^t > h/2$ (respectively, $< -h/2$), molecules are moved one lattice site forward (respectively, backward) along the $x$ direction. If the destination site is occupied, the molecules are not moved, and their position is reset to $x_i^t = h/2$ (respectively, $-h/2$). The same procedure is applied in the $y$ direction. When out-of-equilibrium membrane processes are simulated using Monte-Carlo dynamics, setting the two distinct time-steps required for protein diffusion in the membrane plane and transverse membrane fluctuations is non trivial. This issue is addressed in Ref. [82] (see in particular their Electronic Supplementary Information) and relates to our previous discussion about time-scale separation in Sect. 2. To correctly describe molecular diffusion, the corresponding characteristic time scale must be much larger than the characteristic time scale of membrane fluctuations. In our simulations, the sublattice Langevin dynamics is used to accurately capture the fast-membrane-fluctuation regime. Eq. B.1 shows that the number of MCS between two consecutive jumps of a free molecule can be estimated as $\gamma h^2/k_\mathrm{B} T$. By selecting a sufficiently large value of $\gamma$, we ensure that the particle samples a large-enough number of membrane configurations from an equilibrium distribution before reaching the jump condition. For all the simulations performed, we set $\gamma = 500\,k_\mathrm{B} T/h^2$.

**Insertion:** A site is randomly selected, and if it is empty, a particle is inserted with probability $k_I$. As noted in Ref. [83], the more rigid are the molecules, the lower is their diffusivity. In order to properly compare the results for $\bar{\rho}$ obtained at different $\kappa_1/\kappa_0$ ratios, it is important to ensure that, although $k_D$ is different for each $\kappa_1/\kappa_0$, the dimensionless flux $r = \phi/k_D$ remains the same. This is accomplished by measuring the diffusion rate $k_D^{(t)}$ and the molecule density $\rho^{(t)}$ at each MCS. These values are then used to adjust the insertion rate according to the formula $k_I^{(t)} = r k_D^{(t)}/(1 - \rho^{(t)})$. This procedure guarantees that the dimensionless flux mantains the assigned value $r$. Observe that since one MCS is taken as the time unit, the insertion probability $k_I$ per MCS can be interpreted as an insertion rate. Similarly, the diffusion rate $k_D$ of free molecules—those jumping between two sites lacking occupied nearest neighbors—is determined as the ratio of accepted diffusive jumps.

**Extraction:** If a connected component containing $\geq N_E$ occupied sites is found in the system, all particles in this connected component are removed.

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
