# Peer review of "Molecular sorting on a fluctuating membrane"

_SciPost Physics, doi:SciPost Phys. 18, 099 (2025)_

## Round 1 · Referee Report · Anonymous (Referee 1) · 2024-12-27

Strengths

The manuscript "Molecular sorting on a fluctuating membrane" by Andreghetti and co-authors proposes an extension of previous calculations by the same group, modeling far-from-equilibrium molecular sorting in biological membranes. It is based both on analytical scaling arguments and Monte-Carlo simulations of point-like proteins diffusing on a fluctuating, discretized membrane. The simulations support the analytical findings, at least qualitatively.

The background idea is that attractive forces can lead to protein condensation in small clusters, before they are extracted from the membrane by some active cellular machinery when their size goes above some threshold. The novelty of this work, compared to the previous ones, is that in addition to shorter-range forces of entropic, hydrophobic or molecular origin, it takes into account "long-range" thermal Casimir attraction between proteins, which is well-known to contribute to their condensation in the membrane. As it can be anticipated, the authors demonstrate that taking into account these additional attractive forces facilitates condensation and thus molecular sorting as described above.

Weaknesses

1/ The authors present their results as "new" and they claim that the "role of [Casimir-like interactions] in molecular sorting remains unexplored". Generally speaking, this paper follows a long series of works, some of them dating back over twenty years, and based upon similar physical ingredients (more or less short-range forces, some of them mediated by the membrane, and active membrane recycling) to describe the far-from-equilibrium formation of small protein clusters having some biological function, as clusters. See, e.g., Foret, EPL 71, (2005) ; Truong Quang et al, Curr. Biol. 23, 2197 (2013) ; Fan et al, Phys. Rev. Lett. 104, 118101 (2010) ; or Berger et al, J. Phys. Chem. B 120 10588 (2016). It would be useful to place the present work in this bibliographic context. In order to situate the present model among the previously published ones, the description of the cluster-size distribution obtained through the Monte Carlo simulations could be helpful. It is generally discussed in these anterior works.

2/ When out-of-equilibrium membrane processes are simulated through Monte-Carlo dynamics, it is not trivial to set in the simulations the two different time-steps involved in protein diffusions in the membrane plane on the one side, and transverse membrane fluctuations on the other side. A thorough discussion of this issue has for example recently been proposed in Cornet et al., Soft Matter 20, 4998 (2024).

3/ More critically, the authors do not discuss the order of magnitude of the parameter A in eq 4. According to eq A.10, this parameter remains close to unity when alpha is large. Thus it grows from 0 to ~1 when alpha grows from 1 to large values. For example, A=0.36 if alpha=5. Thus U in eq 4 is at best on the order of kBT. To which extend can an interaction lower than kBT can play a significant role? For example, in eq 10, if |U|<<kBT, U has virtually no effect on Phi. The simulations seem to somewhat contradict what I write here, at least for large values of alpha. Why? Above which value of alpha? Are these values of alpha realistic in the experimental context? Generally speaking, the simulations should also be compared more quantitatively to the analytical findings.

4/ Even more critically: even if U is a bit larger than kBT at short range, it decays rapidly so that the Casimir interaction can effectively be seen as a short-range interaction. Then to what extent its effect is not "trivially" to modify the value of the short-range attraction intensity W? Inspection of figures 2 and 3 indicates that the plots for alpha>1 seem to be essentially just translations of the alpha=1 plot. This supports the hypothesis that the whole effect of the Casimir force can just be absorbed in W.

Report

Several points must be clarified before the paper could be published in Scipost.

Requested changes

1/ Membrane tension (sigma) is ignored. However, it dominates the curvature beyond the length xi=sqrt(kappa/sigma), see e.g. the recent work Fournier, EPJ E 47, 64(2024) and references therein where Casimir forces between proteins are studied in the cell membrane context. xi~10 to 100 nm in plasma membranes. It would be helpful to discuss in the last conclusive section how modifying the long-range shape of U(r) affects (or not) the conclusion of this work.

2/ The calculations leading to eqs 7 and 8 would largely benefit from being detailed, to ease the manuscript reading.

3/ In the legend of figure 2, it would be nice to recall the values of the parameters used in the simulation because they are scattered throughout the paper and somewhat difficult to find. Maybe I’m wrong, but I did not find the value of the lattice spacing used in the simulations, neither the value of the time step (both in real units).

4/ On page 4, the second paragraph compares the diffusion and membrane relaxation time scales, but it is not clear where the result is used later in the manuscript.

Recommendation

Ask for major revision

---

## Round 1 · Referee Report · Anonymous (Referee 2) · 2025-1-7

Report

This article builds on a previously published work (ref. 14 in the manuscript) that deals with the sorting of molecules in biomembranes under the combined effects of molecule adsorption, phase separation into domains and self-extraction of domains above a critical radius. In this paper, the authors study the effect of long-range fluctuation forces (Casimir) between individual molecules and domains (only short-range interactions between molecules were previously taken into account).

The analytical part is based on a calculation of the fluctuation-induced interaction between a molecule, treated as a pointlike modification of the membrane bending and Gaussian rigidity parameters, and a large domain (an aggregate of molecules), assumed close enough to be treated as a semi-infinite plane, and also treated as a region having, with respect to the background membrane, different bending and Gaussian rigidities. The model is well explained and the calculations are sound. Using analytical calculation and numerical calculations, the authors show then that the Casimir-like forces favor the formation of domains and significantly reduce the critical radius for their formation.

In my opinion, this is an interesting work and I recommend its publication in SciPost, provided the authors take into account the following points.

1) Casimir-like forces, due to rigidity contrasts, are not the only membrane-mediated forces. The authors are aware of that and mention this point at the very end of the manuscript. The complete subject would be molecular sorting in the presence of long-range membrane-mediated forces. The authors neglect the intrinsic curvature of the molecules and the intrinsic curvature of the domains, that would yield long-range interactions even at the mean-field level (in the absence of fluctuations). It is acceptable to focus on Casimir forces, especially because they are ubiquitous, but I would prefer to see this important restriction to be stated right in the abstract and the introduction.

2) The physics behind eq. (8) is somewhat hidden. I am guessing that the authors simply perform a first-order Taylor expansion in 1/R with coefficient named R*, and only invoke line-tension as a possible origin ? Is that so, please make this more clear.

3) The domain extraction process is not discussed. Some biophysical background as to how vesicles are formed when R>R_c and how they are extracted would be welcomed. It is actually a complicated process, involving other molecules, that may be depleted, and some intrinsic fluctuations that are neglected here. A word of caution would be welcome.

4) In a real biological membrane, there is a cytoskeleton attached to the membrane. This changes drastically the fluctuation spectrum, so the results obtained in this paper might not apply to biological membranes. The authors could make this statement, probably both in the introduction and in the discussion as a source of possible future investigations.

Recommendation

Publish (meets expectations and criteria for this Journal)

---

## Round 2 · Referee Report · Nicolas Destainville (Referee 1) · 2025-2-25

Strengths
Weaknesses
Report
Recommendation
Publish (easily meets expectations and criteria for this Journal; among top 50%)

---

## Round 2 · Author Response

Strengths
The manuscript ‘‘Molecular sorting on a fluctuating membrane'' by Andreghetti and co-authors proposes an extension of previous calculations by the same group, modeling far-from-equilibrium molecular sorting in biological membranes. It is based both on analytical scaling arguments and Monte-Carlo simulations of point-like proteins diffusing on a fluctuating, discretized membrane. The simulations support the analytical findings, at least qualitatively.
The background idea is that attractive forces can lead to protein condensation in small clusters, before they are extracted from the membrane by some active cellular machinery when their size goes above some threshold. The novelty of this work, compared to the previous ones, is that in addition to shorter-range forces of entropic, hydrophobic or molecular origin, it takes into account ‘‘long-range'' thermal Casimir attraction between proteins, which is well-known to contribute to their condensation in the membrane. As it can be anticipated, the authors demonstrate that taking into account these additional attractive forces facilitates condensation and thus molecular sorting as described above.
## Weaknesses
1/ The authors present their results as ‘‘new'' and they claim that the ‘‘role of [Casimir-like interactions] in molecular sorting remains unexplored". Generally speaking, this paper follows a long series of works, some of them dating back over twenty years, and based upon similar physical ingredients (more or less short-range forces, some of them mediated by the membrane, and active membrane recycling) to describe the far-from-equilibrium formation of small protein clusters having some biological function, as clusters. See, e.g., Foret, EPL 71, (2005); Truong Quang et al, Curr. Biol. 23, 2197 (2013); Fan et al, Phys. Rev. Lett. 104, 118101 (2010); or Berger et al, J. Phys. Chem. B 120 10588 (2016). It would be useful to place the present work in this bibliographic context. In order to situate the present model among the previously published ones, the description of the cluster-size distribution obtained through the Monte Carlo simulations could be helpful. It is generally discussed in these anterior works.
We thank the reviewer for their accurate and insightful comments. A vast body of knowledge has accumulated over the past decades regarding the fluctuational properties of biological membranes. We believe that are our viewpoint on molecular sorting is novel and that, in this context, computing the contribution of Casimir-like interactions is important. At the same time, we agree on the need to more clearly situate our work within the bibliographic context.
Prompted by the reviewer observation, we have added two new paragraphs in the Introduction and in the Conclusions. In the Introduction, we acknowledge that our work is part of a broader line of reasearch on the far-from-equilibrium formation of biomolecular aggregates, and incorporate nine additional references. In the Conclusions, we emphasize that the statistical distribution of cluster sizes is an accessible signature of the underlying aggregation process. We also reference our previous work [15], where we discuss the properties of the cluster-size distribution in our model. The Casimir force studied here does not alter the general shape of this distribution, but shifts the estimated critical radius derived from it, as descrived in Ref. [15] and shown in Fig. 3a.
The following paragraphs have been added in the revised version of our manuscript:
- Introduction, line 39: This perspective fits within the broader framework of the far-from equilibrium formation of biomolecular aggregates with specific functions, including lipid rafts [19,20] and specialized lipid-protein nanodomains [21–25], such as cadherin and integrin clusters [26,27].
- Conclusions, line 321: The statistical distribution of domain sizes provides an accessible signature of many self-organized aggregation processes [24, 26]. The domain size distribution predicted by our model [14] aligns well with experimental data for endocytic sorting and can be used to infer the critical size [15]. This suggests a practical way to experimentally investigate the physical picture of molecular sorting proposed in this work.
2/ When out-of-equilibrium membrane processes are simulated through Monte-Carlo dynamics, it is not trivial to set in the simulations the two different time-steps involved in protein diffusions in the membrane plane on the one side, and transverse membrane fluctuations on the other side. A thorough discussion of this issue has for example recently proposed in Cornet et al., Soft Matter 20, 4998 (2024).
This is indeed a crucial point. We have added a clarifying comment on this issue in the Appendix.
The following paragraphs have been added in the revised version of our manuscript:
- Appendix B, line 433: When out-of-equilibrium membrane processes are simulated using Monte-Carlo dynamics, setting the two distinct time-steps required for protein diffusions in the membrane plane and transverse membrane fluctuations is non trivial. This issue is addressed in Ref. [82] (see in particular their Electronic Supplementing Materials) and relates to our previous discussion about time-scale separation in Sect. 2. To correctly describe molecular diffusion, the corresponding characteristic time scale must be much larger than the characteristic time scale of membrane fluctuations. In our simulations, the sublattice Langevin dynamics for molecules is used to accurately capture the fast-membrane-fluctuation regime. Eq. B.1 shows that the number of MCS between two consecutive jumps of a free molecule can be estimated as $\gamma h^2 / k_{B} T$ . By selecting a sufficiently large value of $\gamma$, we ensure that the particle samples a large-enough number of membrane configurations from an equilibrium distribution before reaching the jump condition. For all the simulations performed, we set $\gamma = 500 k_{B} T / h^2$ .
3/ More critically, the authors do not discuss the order of magnitude of the parameter A in eq 4. According to eq A.10, this parameter remains close to unity when alpha is large. Thus it grows from 0 to $\sim 1$ when alpha grows from 1 to large values. For example, A=0.36 if alpha=5. Thus U in eq 4 is at best on the order of kBT. To which extend can an interaction lower than kBT play a significant role? For example, in eq 10, if $|U| \ll k_{} T$, U has virtually no effect on Phi. The simulations seem to somewhat contradict what I write here, at least for large values of alpha. Why? Above which value of alpha? Are these values of alpha realistic in the experimental context? Generally speaking, the simulations should also be compared more quantitatively to the analytical findings.
To more thoroughly discuss this point, in the revised version of our manuscript we have added new paragraphs and figures (panel (d) in Fig. 3, as well as Figs. 4 and 5 in the Appendix).
The parameter $A$ becomes order of 1 for $\alpha$ in the biologically realistic range $\alpha \sim 10–30$. In that range, $U \gtrsim k_{} T$. The rate of generation of new sorting domains depends exponentially on the critical radius $R_{}$. On the other hand, our Eq. (10) implies that even entropic interactions $\beta U \sim 1$ significantly affect $R_{\mathrm{c}}$. The entropic force is stronger at short distances, where it facilitates nucleation by biasing molecular diffusion toward sorting domains and stabilizing them. In the present out-of-equilibrium setting, there isn't a critical value of $\alpha$ above which demixing occurs; instead, the nucleation rate of sorting domains increases smoothly as a function of the inclusions' rigidity.
Additionally, in the Appendix we have included the result of a perturbative computation (whose details were omitted for brevity) showing that at short separations, $U$ is much larger than its estimate given by Eq. 4, which therefore provides only a lower bound on $U$.
As an aside, we have also slightly generalized our previous computation to account for the possibility that the molecule and domain have different rigidities, $\kappa _{2}$ and $\kappa _{1}$, respectively.
The following paragraphs and figures have been added in the revised version of our manuscript:
- Sect. 2.2, line162: For realistic values $\alpha \sim 10 - 30$ [47–49], the potential $U$, induced by membrane fluctuations, is at least of the order of $k_{\mathrm{B}} T$ when $r \sim R$ and tends to zero as $r$ grows (see Appendix A).
- Sect. 2.2, line 197: The following physical picture thus emerges. The efficiency of the sorting process is controlled by the rate of nucleation of new sorting domains. According to classical nucleation theory the rate of generation of new sorting domains depends exponentially on $R_{\mathrm{c}}$ [52]. Eq. (10) implies that even entropic interactions $\beta U \sim 1$ significantly affect $R_{\mathrm{c}}$. At short distances, the entropic force acts as a facilitator of nucleation by biasing molecular diffusion toward sorting domains and stabilizing them. While at equilibrium a sharp demixing transition is observed above a critical value of rigidity [46], in the statistical steady state of interest here we expect to observe a smooth increase of the rate of nucleation of sorting domains with increasing rigidity of the inclusions. A significant effect is expected in particular in the realistic range $\alpha \sim 10 - 30$ [47–49], where $\beta U \sim 1$ in the proximity of the domains.
- Sect. 3, line 278: As expected, the effect of entropic forces on the sorting process exhibits a smooth dependence on the relative rigidity α, and it becomes particularly significant for $\alpha \gtrsim 10$, where$\beta U \sim 1$ (Fig. 3d).
- Sect. 3, figure: We have added panel d) in Fig. 3, which illustrates the smooth decrease in individual molecule density with increasing relative rigidity $\alpha$, becoming particularly significant (larger than 50% reduction) for $\alpha \gtrsim 10$.
- Appendix A, line 377, formula with generic $\kappa _{2}$:
\
When the molecule and the domain have the same rigidity ($\kappa _{2} = \kappa _{1}$), we obtain * Appendix A, line 380: which is a monotonically increasing function for $\alpha > 1$, taking on values of order 1 for $\alpha \gtrsim 10$ (Fig. 4). * Appendix A, line 396: In the previous paragraphs, we assumed that the distance $x$ between molecule and domain was much larger than the size $a$ of the molecule. Here, we take into account the finite size of the inclusion by modeling it as a disk of rigidity $\kappa _{2}$ and radius $a$ centered in the point of coordinates $(x, 0)$:
\
The resulting interaction energy can be computed as
\begin{eqnarray} U_{\mathrm{Disk}} & = & - k_{\mathrm{B}} T \log \frac{Z}{Z_{0}} \hspace{0.27em} \hspace{0.27em} = \hspace{0.27em} \hspace{0.27em} - k_{\mathrm{B}} T \sum_{n = 1}^{\infty} \frac{(- \beta)^n}{n!} \langle \lbrace \delta \mathcal{H}_{\mathrm{Disk}} \rbrace^n \rangle _{0, \mathrm{c}} \nonumber \end{eqnarray}
where $\langle \cdots \rangle _{\mathrm{c}}$ denotes connected averages. Resummation of similar perturbation series has been performed up to finite orders in analogous geometries, either through direct computation of Feynman diagrams [45] or via numerical methods [44], demonstrating that the interaction energy increases sharply at short separations. We have evaluated both $U$ from Eq. A.9 and $U_{\mathrm{Disk}}$ from Eq. A.16 at first order in $(\kappa _{2} - \kappa _{0}) / \kappa _{0}$, getting
\begin{eqnarray} \frac{U_{\mathrm{Disk}}^{(1)}}{U^{(1)}} & = & 2 \left[ \frac{1}{\sqrt{1 - (a / x)^2}} - 1 \right] \left( \frac{x}{a} \right)^2 \end{eqnarray}
This relative finite-size correction is plotted in Fig. 5. It shows that Eq. A.9, which neglects finite-size effects, significantly underestimates the interaction energy at short distances while accurately capturing its behaviour at length scales much larger than the inclusion size. * Appendix A, figures: New Fig. 4, showing the behavior of the parameter $A$ as a function of the relative rigidity $\alpha$, and new Fig. 5, showing the finite-size correction described in the text.
4/ Even more critically: even if U is a bit larger than kBT at short range, it decays rapidly so that the Casimir interaction can effectively be seen as a short-range interaction. Then to what extent its effect is not ‘‘trivially'' to modify the value of the short-range attraction intensity W? Inspection of figures 2 and 3 indicates that the plots for alpha>1 seem to be essentially just translations of the alpha=1 plot. This supports the hypothesis that the whole effect of the Casimir force can just be absorbed in W.
We agree with the Reviewer's assessment: the Casimir interaction is stronger at short separations and, in terms of the macroscopic kinetics, effectively renormalizes the short-range attraction strength $W$. Several key points are worth emphasizing here: a) The Casimir force has a physical origin and provides a universal mechanism contributing to molecular aggregation. b) It depends on a physical parameter—the relative rigidity of the inclusions—which can be tuned experimentally. c) In the presence of weak direct intermolecular interactions, the Casimir contribution may be crucial for efficient sorting.
We have extended the discussion on these points in both the main text and in the Conclusions.
It is worth observing that the Casimir interaction, although technically long-range, is mainly relevant at short separations. To prevent possible misunderstandings, we have revised our terminology: we now use “direct” (non membrane-mediated) instead of “short range” and have avoided the term “long-range”.
The following paragraphs have been added in the revised version of our manuscript:
- Sect. 3, line 264: Since the entropic force mainly acts at short separations, it renormalizes the value of the direct interaction, resulting in an effective short-range interaction strength $W_{\operatorname{eff}}$, as evidenced by the consistent shift of the density curves toward lower values of W in Fig. 2. However, it is important to note that the entropic component of $W_{\operatorname{eff}}$ has a distinct origin and parametric dependence compared to the direct interaction part, as it is governed by the relative rigidity $\alpha$.
- Fig. 2, caption: For larger values of the relative rigidity α, the density curve and the optimal interaction strength $W_{\operatorname{opt}}$ shift toward lower values of $W$.
- Conclusions, line 298: Note that from the point of view of macroscopic kinetics, the entropic forces can be described by a short-range interaction of the same type as in Eq. (11), with an effective parameter $W_{\operatorname{eff}}$ , corresponding to attraction of particles toward the domains. The value of $W_{\operatorname{eff}}$ is determined by the relative rigidity of the domain and has order $k_{B} T$. As a result, at low and moderate values of $W$, entropic forces provide a universal physical mechanism for the aggregation of molecules within the membrane, independent of microscopic interaction details. We believe that this should be taken into account in the design and interpretation of biological experiments.
## Report
Several points must be clarified before the paper could be published in Scipost.
## Requested changes
1/ Membrane tension (sigma) is ignored. However, it dominates the curvature beyond the length xi=sqrt(kappa/sigma), see e.g. the recent work Fournier, EPJ E 47, 64 (2024) and references therein where Casimir forces between proteins are studied in the cell membrane context. xi~10 to 100 nm in plasma membranes. It would be helpful to discuss in the last conclusive section how modifying the long-range shape of U(r) affects (or not) the conclusion of this work.
We thank the reviewer for their accurate and insightful comments. We have now included a more detailed discussion about this point.
The following paragraphs have been added in the revised version of our manuscript:
- Sect. 2.1, line 138: Notice that when considering a membrane surface tension $\sigma$, a new relevant lengthscale, $\xi \sim \sqrt{\kappa / \sigma}$, emerges [65–67]. At scales below $\xi$, surface tension has a weak influence on membrane properties, whereas for scales above $\xi$, it significantly modifies the long-range part of the entropic interaction [44, 65–67]. As discussed in Refs. [44, 45] and in Appendix A, the entropic interaction is mainly appreciable at short separations. Therefore, we expect the effects of surface tension to be negligible in the present context.
- Sect. 3, line 280: Furthermore, we tested the effect of including a
surface term in our simulations and observed no significant change up to the
realistic value $\sigma = 10^{\nonconverted{minus} 5} \,
^{\nonconverted{minus} 2}$ [74], in line with theoretical arguments.
2/ The calculations leading to eqs 7 and 8 would largely benefit from being detailed, to ease the manuscript reading.
We have added more details on the steps leading to eqs. 7 and 8.
The following paragraphs have been included in the revised version of our manuscript:
- Sect. 22, line 164: The potential $U (r)$ rapidly approaches zero when $r$ becomes much larger than $R$ (as $\sim (r / R)^{- 4}$, see Eq. (A.14)). This allows us to neglect $U (r)$ in Eq. (6) when $r \gg R$. The leading contribution in $r / R$ can be extracted by integrating by parts in the integral in Eq. (6):
\
where $\delta \mathcal{J}$ converges as $r \to \infty$:
\
Since $| \delta \mathcal{J} | \leqslant A \hspace{0.17em} a / R \ll 1$ for $a \ll R$, it can be neglected, leading to the relation * Sect. 2.2, line 172: using the Gibbs-Thomson relation [70,71],
3/ In the legend of figure 2, it would be nice to recall the values of the parameters used in the simulation because they are scattered throughout the paper and somewhat difficult to find. Maybe I'm wrong, but I did not find the value of the lattice spacing used in the simulations, neither the value of the time step (both in real units).
We have added information about the lattice spacing and time step, and summarized the values of all parameters used in the simulation in the legend of Fig. 2.
The following paragraphs have been included in the revised version of our manuscript:
- Sect. 3, line 245: To match simulation parameters with real-world units, we consider that each square plaquette in the lattice corresponds to a patch of lipids of area $h^2$, with $h \approx 10\operatorname{nm}$, the order of magnitude of the lateral size of typical protein inclusions, and also of the shortest fluctuational wavelengths for membrane bending deformations [46, 74]. For molecular diffusivities $D \approx 1 ^2 / $, the typical time between consecutive diffusive jumps of a free inclusion on the lattice is $k^{\nonconverted{minus} 1}_{D} = h^2 / D \approx 10^{\nonconverted{minus} 4} $.
- Fig. 2, caption: Simulations were performed with $\phi / k_{\mathrm{D}} = 10^{- 5}$, $\kappa _{0} = - \bar{\kappa}_{0} = 10 \hspace{0.17em} k_{\mathrm{B}} T$, $\kappa _{1} = - \bar{\kappa}_{1}$, $L = 100$, $N_{\mathrm{E}} = 25$. For $h = 10 \hspace{0.17em}$nm and $D = 1 \hspace{0.17em} \mu \mathrm{m}^2 / \mathrm{s}$, one has $k_{D}^{- 1} = 10^{- 4} \hspace{0.17em}$s.
4/ On page 4, the second paragraph compares the diffusion and membrane relaxation time scales, but it is not clear where the result is used later in the manuscript.
We have now added a cross-reference to that paragraph, clarifying that the discussed timescale separation is needed to justify the adiabatic approximation used in the following sections. The argument is now recalled again in the Appendix to better explain how distinct time steps are chosen for, respectively, protein diffusion and membrane fluctuations in the numerical method.
The following paragraphs have been included in the revised version of our manuscript:
- Sect. 2.1, line 126: In the adiabatic approximation, justified by the timescale separation $\tau _{\operatorname{diff}} \gg \tau _{\operatorname{rel}}$ , molecules included within the membrane experience effective forces that can be computed by averaging over membrane fluctuations sampled from the equilibrium distribution.
- Appendix, line 433: When out-of-equilibrium membrane processes are simulated using Monte-Carlo dynamics, setting the two distinct time-steps required for protein diffusion in the membrane plane and transverse membrane fluctuations is non trivial. This issue is addressed in Ref. [82] (see in particular their Electronic Supplementary Information) and relates to our previous discussion about time-scale separation in Sect. 2. To correctly describe molecular diffusion, the corresponding characteristic time scale must be much larger than the characteristic time scale of membrane fluctuations.
## Report #2 by Anonymous (Referee 1) on 2025-1-7 (Invited Report)**
## Report
This article builds on a previously published work (ref. 14 in the manuscript) that deals with the sorting of molecules in biomembranes under the combined effects of molecule adsorption, phase separation into domains and self-extraction of domains above a critical radius. In this paper, the authors study the effect of long-range fluctuation forces (Casimir) between individual molecules and domains (only short-range interactions between molecules were previously taken into account).
The analytical part is based on a calculation of the fluctuation-induced interaction between a molecule, treated as a pointlike modification of the membrane bending and Gaussian rigidity parameters, and a large domain (an aggregate of molecules), assumed close enough to be treated as a semi-infinite plane, and also treated as a region having, with respect to the background membrane, different bending and Gaussian rigidities. The model is well explained and the calculations are sound. Using analytical calculation and numerical calculations, the authors show then that the Casimir-like forces favor the formation of domains and significantly reduce the critical radius for their formation.
In my opinion, this is an interesting work and I recommend its publication in SciPost, provided the authors take into account the following points.
1) Casimir-like forces, due to rigidity contrasts, are not the only membrane-mediated forces. The authors are aware of that and mention this point at the very end of the manuscript. The complete subject would be molecular sorting in the presence of long-range membrane-mediated forces. The authors neglect the intrinsic curvature of the molecules and the intrinsic curvature of the domains, that would yield long-range interactions even at the mean-field level (in the absence of fluctuations). It is acceptable to focus on Casimir forces, especially because they are ubiquitous, but I would prefer to see this important restriction to be stated right in the abstract and the introduction.
We have followed the reviewer's suggestion. We now mention directly in the Abstract that our work addresses Casimir-like forces between rigid membrane inclusions, and further explain in the Introduction that while membrane-mediated interactions can originate either from the presence of protein inclusions with intrinsic curvature or from protein inclusions with higher rigidity, our work focuses on the latter case of interactions.
The following changes have been included in the revised version of our manuscript:
- Abstract: Casimir-like forces between rigid membrane inclusions
- Introduction, line 46: Moreover, it has long been established that protein inclusions in lipid membranes are subject to membrane-mediated interactions. These can originate either from ground state deformation of membrane shape, when protein inclusions are a source of intrinsic curvature, or from membrane fluctuations, as the presence of embedded protein inclusions restricts membrane fluctuation modes, generating entropic interactions [40–42]. We focus here on the latter class of interactions, commonly known as Casimir-like forces. These are non-additive, weak forces that are mainly relevant at short separations [43–45]. At thermodynamic equilibrium, they are however sufficient to induce a demixing transition in heterogeneous membranes [46].
2) The physics behind eq. (8) is somewhat hidden. I am guessing that the authors simply perform a first-order Taylor expansion in 1/R with coefficient named R*, and only invoke line-tension as a possible origin ? Is that so, please make this more clear.
We have added more information explaining that eq. (8) is a form of the Gibbs-Thomson relation. In parallel, we have also provided more details on the derivation of eq. (7).
The following paragraphs have been included in the revised version of our manuscript:
- Sect. 2.2, line 172: using the Gibbs-Thomson relation [70,71],
- Sect. 22, line 168: The potential $U (r)$ rapidly approaches zero when $r$ becomes much larger than $R$ (as $\sim (r / R)^{- 4}$, see Eq. (A.14)). This allows us to neglect $U (r)$ in Eq. (6) when $r \gg R$. The leading contribution in $r / R$ can be extracted by integrating by parts in the integral in Eq. (6):
\
where $\delta \mathcal{J}$ converges as $r \to \infty$:
\
Since $\delta \mathcal{J} \sim A \hspace{0.17em} (a / R)^2 \ll 1$ for $a \ll R$, it can be neglected, leading to the relation
3) The domain extraction process is not discussed. Some biophysical background as to how vesicles are formed when R>R_c and how they are extracted would be welcomed. It is actually a complicated process, involving other molecules, that may be depleted, and some intrinsic fluctuations that are neglected here. A word of caution would be welcome.
As mentioned by the reviewer, the actual biophysical process of vesicle formation and extraction is quite complex. We have added a word of caution on this point, and supported it with a reference to a comprehensive and authoritative review on the topic.
The following change has been included in the revised version of our manuscript:
- Sect. 2, line 71: It is worth observing here that vesicle formation is a complex process involving the concomitant action of a wide variety of genes, as reviewed, for instance, in Ref. [1]. In our approach, we abstract on molecular details and encode the mesoscopic effect of vesicle extraction in the single parameter $R_{}$.
4) In a real biological membrane, there is a cytoskeleton attached to the membrane. This changes drastically the fluctuation spectrum, so the results obtained in this paper might not apply to biological membranes. The authors could make this statement, probably both in the introduction and in the discussion as a source of possible future investigations.
We have added a discussion on this topic in the conclusion.
The following change has been included in the revised version of our manuscript:
- Conclusion, line 310: Here, we did not account for the role of the cytoskeleton, which could influence the membrane fluctuation spectrum. At the mesoscopic scale, a more accurate description can be achieved by incorporating additional terms into the membrane Hamiltonian (Eq. 3) to account for interactions with the cytoskeletal network. Previous research has explored this aspect by introducing local membrane pinning [77, 78] or by considering membrane confinement [79]. These studies found that at short wavelengths, the fluctuation spectrum of a free membrane is retrieved. This suggests that the effect of the cytoskeleton on entropic forces, which in the present context mainly act at short separations, may be weak. This point will be the subject of future investigations.

---

## Round 2 · List of Changes

**Abstract:** Casimir-like forces between rigid membrane inclusions
**Introduction, line 39:** This perspective fits within the broader framework
of the far-from equilibrium formation of biomolecular aggregates with specific
functions, including lipid rafts [19,20] and specialized lipid-protein
nanodomains [21–25], such as cadherin and integrin clusters [26,27].
**Introduction, line 46:** Moreover, it has long been established that protein
inclusions in lipid membranes are subject to membrane-mediated interactions.
These can originate either from ground state deformation of membrane shape,
when protein inclusions are a source of intrinsic curvature, or from membrane
fluctuations, as the presence of embedded protein inclusions restricts
membrane fluctuation modes, generating entropic interactions [40–42]. We focus
here on the latter class of interactions, commonly known as Casimir-like
forces. These are non-additive, weak forces that are mainly relevant at short
separations [43–45]. At thermodynamic equilibrium, they are however sufficient
to induce a demixing transition in heterogeneous membranes [46].
**Sect. 2, line 71:** It is worth observing here that vesicle formation is a
complex process involving the concomitant action of a wide variety of genes,
as reviewed, for instance, in Ref. [1]. In our approach, we abstract on
molecular details and encode the mesoscopic effect of vesicle extraction in
the single parameter $R\_{}$.
**Sect. 2.1, line 126:** In the adiabatic approximation, justified by the
timescale separation $\tau \_{\operatorname{diff}} \gg \tau
\_{\operatorname{rel}}$ , molecules included within the membrane experience
effective forces that can be computed by averaging over membrane fluctuations
sampled from the equilibrium distribution.
**Sect. 2.1, line 138:** Notice that when considering a membrane surface
tension $\sigma$, a new relevant lengthscale, $\xi \sim \sqrt{\kappa /
\sigma}$, emerges [65–67]. At scales below $\xi$, surface tension
has a weak influence on membrane properties, whereas for scales
above $\xi$, it significantly modifies the long-range part of the
entropic interaction [44, 65–67]. As discussed in Refs. [44, 45] and in
Appendix A, the entropic interaction is mainly appreciable at short
separations. Therefore, we expect the effects of surface tension to be
negligible in the present context.
**Sect. 2.2, line162:** For realistic values $\alpha \sim 10 -
30$ [47–49], the potential $U$, induced by membrane fluctuations, is at
least of the order of $k\_{\mathrm{B}} T$ when $r \sim R$ and tends to zero as
$r$ grows (see Appendix A).
**Sect. 22, line 164:** The potential $U (r)$ rapidly approaches zero
when $r$ becomes much larger than $R$ (as $\sim (r / R)^{- 4}$, see
Eq. (A.14)). This allows us to neglect $U (r)$ in Eq. (6) when $r \gg R$.
The leading contribution in $r / R$ can be extracted by integrating by parts
in the integral in Eq. (6):
\\[ \mathcal{J}= \int\_{R}^r \frac{\mathrm{d} \rho}{\rho} \mathrm{e}^{\beta U
(\rho)} = \ln \frac{r}{R} + \delta \mathcal{J}, \\]
where $\delta \mathcal{J}$ converges as $r \to \infty$:
\\[ \delta \mathcal{J} \approx - \beta \int\_{R}^{\infty} \mathrm{d} \rho \,
\frac{\mathrm{d} U (\rho)}{\mathrm{d} \rho} \ln \left( \frac{\rho}{R}
\right) \mathrm{e}^{\beta U (\rho)} . \\]
Since $| \delta \mathcal{J} | \leqslant A \hspace{0.17em} a / R \ll 1$ for $a
\ll R$, it can be neglected, leading to the relation
**Sect. 22, line 168:** The potential $U (r)$ rapidly approaches zero when $r$
becomes much larger than $R$ (as $\sim (r / R)^{- 4}$, see Eq. (A.14)).
This allows us to neglect $U (r)$ in Eq. (6) when $r \gg R$. The leading
contribution in $r / R$ can be extracted by integrating by parts in the
integral in Eq. (6):
\\[ \mathcal{J}= \int\_{R}^r \frac{\mathrm{d} \rho}{\rho} \mathrm{e}^{\beta U
(\rho)} = \ln \frac{r}{R} + \delta \mathcal{J}, \\]
where $\delta \mathcal{J}$ converges as $r \to \infty$:
\\[ \delta \mathcal{J} \approx - \beta \int\_{R}^{\infty} \mathrm{d} \rho
\frac{\mathrm{d} U (\rho)}{\mathrm{d} \rho} \ln \left( \frac{\rho}{R}
\right) \mathrm{e}^{\beta U (\rho)} . \\]
Since $\delta \mathcal{J} \sim A \hspace{0.17em} (a / R)^2 \ll 1$ for $a \ll
R$, it can be neglected, leading to the relation
**Sect. 2.2, line 172:** using the Gibbs-Thomson relation [70,71],
**Sect. 2.2, line 197:** The following physical picture thus emerges. The
efficiency of the sorting process is controlled by the rate of nucleation of
new sorting domains. According to classical nucleation theory the rate of
generation of new sorting domains depends exponentially on
$R\_{\mathrm{c}}$ [52]. Eq. (10) implies that even entropic
interactions $\beta U \sim 1$ significantly affect $R\_{\mathrm{c}}$. At
short distances, the entropic force acts as a facilitator of nucleation by
biasing molecular diffusion toward sorting domains and stabilizing them. While
at equilibrium a sharp demixing transition is observed above a critical value
of rigidity [46], in the statistical steady state of interest here we
expect to observe a smooth increase of the rate of nucleation of sorting
domains with increasing rigidity of the inclusions. A significant effect
is expected in particular in the realistic range $\alpha \sim 10 -
30$ [47–49], where $\beta U \sim 1$ in the proximity of the domains.
**Sect. 3, line 245:** To match simulation parameters with real-world units,
we consider that each square plaquette in the lattice corresponds to a patch
of lipids of area $h^2$, with $h \approx 10\operatorname{nm}$, the order of
magnitude of the lateral size of typical protein inclusions, and also of the
shortest fluctuational wavelengths for membrane bending deformations [46, 74].
For molecular diffusivities $D \approx 1 ^2 / $, the typical time between
consecutive diffusive jumps of a free inclusion on the lattice is
$k^{\nonconverted{minus} 1}\_{D} = h^2 / D \approx 10^{\nonconverted{minus} 4}
$.
**Sect. 3, line 264:** Since the entropic force mainly acts at short
separations, it renormalizes the value of the direct interaction, resulting in
an effective short-range interaction strength $W\_{\operatorname{eff}}$, as
evidenced by the consistent shift of the density curves toward lower values of
W in Fig. 2. However, it is important to note that the entropic component of
$W\_{\operatorname{eff}}$ has a distinct origin and parametric dependence
compared to the direct interaction part, as it is governed by the relative
rigidity $\alpha$.
**Sect. 3, line 278:** As expected, the effect of entropic forces on the
sorting process exhibits a smooth dependence on the relative rigidity α,
and it becomes particularly significant for $\alpha \gtrsim 10$, where$\beta U
\sim 1$ (Fig. 3d).
**Sect. 3, line 280:** Furthermore, we tested the effect of including a
surface term in our simulations and observed no significant change up to the
realistic value $\sigma = 10^{\nonconverted{minus} 5} \,
^{\nonconverted{minus} 2}$ [74], in line with theoretical arguments.
**Conclusions, line 298:** Note that from the point of view of macroscopic
kinetics, the entropic forces can be described by a short-range interaction of
the same type as in Eq. (11), with an effective parameter
$W\_{\operatorname{eff}}$ , corresponding to attraction of particles toward
the domains. The value of $W\_{\operatorname{eff}}$ is determined by the
relative rigidity of the domain and has order $k\_{B} T$. As a result, at low
and moderate values of $W$, entropic forces provide a universal physical
mechanism for the aggregation of molecules within the membrane, independent of
microscopic interaction details. We believe that this should be taken into
account in the design and interpretation of biological experiments.
**Conclusion, line 310:** Here, we did not account for the role of the
cytoskeleton, which could influence the membrane fluctuation spectrum. At the
mesoscopic scale, a more accurate description can be achieved by incorporating
additional terms into the membrane Hamiltonian (Eq. 3) to account for
interactions with the cytoskeletal network. Previous research has explored
this aspect by introducing local membrane pinning [77, 78] or by considering
membrane confinement [79]. These studies found that at short wavelengths, the
fluctuation spectrum of a free membrane is retrieved. This suggests that the
effect of the cytoskeleton on entropic forces, which in the present context
mainly act at short separations, may be weak. This point will be the subject
of future investigations.
**Conclusions, line 321:** The statistical distribution of domain sizes
provides an accessible signature of many self-organized aggregation processes
[24, 26]. The domain size distribution predicted by our model [14] aligns well
with experimental data for endocytic sorting and can be used to infer the
critical size [15]. This suggests a practical way to experimentally
investigate the physical picture of molecular sorting proposed in this work.
**Appendix A, line 377, formula with generic $\kappa \_{2}$:**
\\[ U (x) = - k\_{B} T \frac{(\kappa \_{1} - \kappa \_{0})}{4 (\kappa \_{1} +
3
\kappa \_{0})} \left( \frac{\kappa \_{2} - \kappa \_{0}}{\kappa \_{2} + 3
\kappa \_{0}} \right) \left[ \frac{15 \kappa \_{2} \kappa \_{1} + 13 \kappa
\_{2} \kappa \_{0} + 21 \kappa \_{0} \kappa \_{1} + 15 \kappa
\_{0}^2}{(\kappa \_{2} + \kappa \_{0}) (\kappa \_{0} + 3 \kappa \_{1})}
\right] \frac{a^2}{x^2} \\]
When the molecule and the domain have the same rigidity ($\kappa \_{2} =
\kappa \_{1}$), we obtain
**Appendix A, line 380:** which is a monotonically increasing function for
$\alpha > 1$, taking on values of order 1 for $\alpha \gtrsim 10$
(Fig. 4).
**Appendix A, line 396:** In the previous paragraphs, we assumed that the
distance $x$ between molecule and domain was much larger than the size $a$ of
the molecule. Here, we take into account the finite size of the inclusion by
modeling it as a disk of rigidity $\kappa \_{2}$ and radius $a$ centered in
the point of coordinates $(x, 0)$:
\\[ \delta \mathcal{H}\_{\mathrm{Disk}} = \frac{(\kappa \_{2} - \kappa
\_{0})}{2} \int\_{\mathrm{Disk}} d \mathbf{r} \lbrace (\nabla^2 u
(\mathbf{r}))^2 - 2 [\partial \_{x}^2 u (\mathbf{r}) \partial \_{y}^2 u
(\mathbf{r}) - \partial \_{x} \partial \_{y} u (\mathbf{r}) \partial \_{x}
\partial \_{y} u (\mathbf{r})] \rbrace \\]
The resulting interaction energy can be computed as
\begin{eqnarray}
U\_{\mathrm{Disk}} & = & - k\_{\mathrm{B}} T \log \frac{Z}{Z\_{0}}
\hspace{0.27em} \hspace{0.27em} = \hspace{0.27em} \hspace{0.27em} -
k\_{\mathrm{B}} T \sum\_{n = 1}^{\infty} \frac{(- \beta)^n}{n!} \langle
\lbrace \delta \mathcal{H}\_{\mathrm{Disk}} \rbrace^n \rangle \_{0,
\mathrm{c}} \nonumber
\end{eqnarray}
where $\langle \cdots \rangle \_{\mathrm{c}}$ denotes connected averages.
Resummation of similar perturbation series has been performed up to finite
orders in analogous geometries, either through direct computation of Feynman
diagrams [45] or via numerical methods [44], demonstrating that the
interaction energy increases sharply at short separations. We have evaluated
both $U$ from Eq. A.9 and $U\_{\mathrm{Disk}}$ from Eq. A.16 at
first order in $(\kappa \_{2} - \kappa \_{0}) / \kappa \_{0}$, getting
\begin{eqnarray*}
\frac{U\_{\mathrm{Disk}}^{(1)}}{U^{(1)}} & = & 2 \left[ \frac{1}{\sqrt{1 -
(a / x)^2}} - 1 \right] \left( \frac{x}{a} \right)^2
\end{eqnarray*}
This relative finite-size correction is plotted in Fig. 5. It shows that
Eq. A.9, which neglects finite-size effects, significantly underestimates
the interaction energy at short distances while accurately capturing its
behaviour at length scales much larger than the inclusion size.
**Appendix B, line 433:** When out-of-equilibrium membrane processes are
simulated using Monte-Carlo dynamics, setting the two distinct time-steps
required for protein diffusions in the membrane plane and transverse membrane
fluctuations is non trivial. This issue is addressed in Ref. [82] (see in
particular their Electronic Supplementing Materials) and relates to our
previous discussion about time-scale separation in Sect. 2. To correctly
describe molecular diffusion, the corresponding characteristic time scale must
be much larger than the characteristic time scale of membrane fluctuations.
In our simulations, the sublattice Langevin dynamics for molecules is used to
accurately capture the fast-membrane-fluctuation regime. Eq. B.1 shows that
the number of MCS between two consecutive jumps of a free molecule can be
estimated as $\gamma h^2 / k\_{B} T$ . By selecting a sufficiently large value
of $\gamma$, we ensure that the particle samples a large-enough number of
membrane configurations from an equilibrium distribution before reaching the
jump condition. For all the simulations performed, we set $\gamma = 500 k\_{B}
T / h^2$ .
**Fig. 2, caption:** For larger values of the relative rigidity α, the
density curve and the optimal interaction strength $W\_{\operatorname{opt}}$
shift toward lower values of $W$.
**Fig. 2, caption:** Simulations were performed with $\phi / k\_{\mathrm{D}} =
10^{- 5}$, $\kappa \_{0} = - \bar{\kappa}\_{0} = 10 \hspace{0.17em}
k\_{\mathrm{B}} T$, $\kappa \_{1} = - \bar{\kappa}\_{1}$, $L = 100$,
$N\_{\mathrm{E}} = 25$. For $h = 10 \hspace{0.17em}$nm and $D = 1
\hspace{0.17em} \mu \mathrm{m}^2 / \mathrm{s}$, one has $k\_{D}^{- 1} = 10^{-
4} \hspace{0.17em}$s.
**Sect. 3, figure:** We have added panel d) in Fig. 3, which illustrates the
smooth decrease in individual molecule density with increasing relative
rigidity $\alpha$, becoming particularly significant (larger than 50%
reduction) for $\alpha \gtrsim 10$.
**Appendix A, figures:** New Fig. 4, showing the behavior of the parameter $A$
as a function of the relative rigidity $\alpha$, and new Fig. 5, showing the
finite-size correction described in the text.

---

## Editorial Decision

published